# Advances in Focused Ion Beam Tomography for Three-Dimensional Characterization in Materials Science

**DOI:** 10.3390/ma16175808

**Published:** 2023-08-24

**Authors:** Francesco Mura, Flavio Cognigni, Matteo Ferroni, Vittorio Morandi, Marco Rossi

**Affiliations:** 1Department of Basic and Applied Sciences, University of Rome “La Sapienza”, Via Antonio Scarpa 14, 00161 Rome, Italy; flavio.cognigni@uniroma1.it (F.C.); marco.rossi@uniroma1.it (M.R.); 2National Research Council of Italy, Institute for Microelectronics and Microsystems, Section of Bologna, Via Piero Gobetti 101, 40129 Bologna, Italy; matteo.ferroni@unibs.it (M.F.); morandi@bo.imm.cnr.it (V.M.); 3Department of Civil, Environmental, Architectural Engineering and Mathematics (DICATAM), University of Brescia, Via Branze 43, 25123 Brescia, Italy

**Keywords:** FIB-SEM tomography, 3D reconstruction, porous material systems, segmentation

## Abstract

Over the years, FIB-SEM tomography has become an extremely important technique for the three-dimensional reconstruction of microscopic structures with nanometric resolution. This paper describes in detail the steps required to perform this analysis, from the experimental setup to the data analysis and final reconstruction. To demonstrate the versatility of the technique, a comprehensive list of applications is also summarized, ranging from batteries to shale rocks and even some types of soft materials. Moreover, the continuous technological development, such as the introduction of the latest models of plasma and cryo-FIB, can open the way towards the analysis with this technique of a large class of soft materials, while the introduction of new machine learning and deep learning systems will not only improve the resolution and the quality of the final data, but also expand the degree of automation and efficiency in the dataset handling. These future developments, combined with a technique that is already reliable and widely used in various fields of research, are certain to become a routine tool in electron microscopy and material characterization.

## 1. Introduction

Over the last decades, transmission electron microscopy (TEM) and scanning electron microscopy (SEM) have become essential analytical techniques in both the materials and life sciences. Electron microscopy complements optical microscopy because of its higher spatial resolution and the information provided by the peculiar interaction between the electron beam and the specimen. In addition, modern microscopes are equipped with additional analytical accessories, such as X-ray spectroscopy (both wavelength and energy dispersion), electron backscatter diffraction (EBSD), Auger electron spectroscopy (AE), or cathodoluminescence (CL), to allow a deeper characterization of the sample and to provide further information on the chemical content of the samples. Although instrumental development has brought resolution to the nanometric level for SEM and even beyond for TEM [1], electron microscopy remains essentially a visualization technique, where the spatial arrangement of the constituents is represented in a two-dimensional image. The possibility of a three-dimensional (3D) reconstruction in the micrometer range required the long-awaited integration of computer-assisted methods with the imaging capability of electron microscopes. The first approaches consisted of the tomographic elaboration of a series of TEM projective images taken at different angles [2] or the combination of a few conventional SEM images into a so-called photogrammetry visualization [3]. The limitations of both techniques are obvious: TEM tomography is limited to very small volumes because the sample has to be very thin (50–100 nm of thickness) in order to be electron transparent, producing a complex sample preparation; reconstruction by SEM photogrammetry is limited to the external shape of the object, with no information about the inner structure, although the possibility of performing scanning transmission electron microscopy (STEM) inside a SEM seems very promising for the performance of successful electron tomography [4]. This is due to the internal structure of a SEM, which facilitates the rotation of the holder and, in the absence of a post-specimen lens, allows an almost complete collection of the transmitted electrons [5].

The fundamental advance in the 3D approach in the SEM platform came with the introduction of an electromagnetic column capable of accelerating, focusing, and rastering a beam of positively charged particles. Focused ion beam (FIB) systems were originally developed to exploit the ability of the energetic ion beam to erode the sample surface as a tool to expose subsurface areas of the sample for secondary ion mass spectroscopy [6]. Currently, ion beams are generated by liquid metal or ionized plasma sources [7], with Ga^+^ ions being the dominant technology, and FIB can ablate materials from a micrometric range with adequate accuracy. In addition to the milling capability, the emission of secondary electrons generated by the ion–sample interaction provides a signal that can be used by the detection system of the SEM to form an ion beam microscopic image. For these reasons, FIB technology has rapidly advanced in performance and integration with the SEM, becoming the so-called “dual FIB-SEM system”, which is capable of direct control of the ablation process through simultaneous FIB or SEM imaging. The 3D reconstruction approach in the SEM is based on the regular alternation of ion milling and visualization of the exposed section of the sample. This repetition of milling and imaging, commonly referred to as “slice and view”, generates a sequence of images that will form the input for a computer-assisted 3D digital reconstruction of the sample volume. In fact, tomography stands for “writing by cutting” and is a method of fully representing an object.

Figure 1, edited from the one compiled by Cantoni and Holzer [8], compares the main 3D tomographic techniques based on X-ray and nano-X-ray imaging, mechanical serial sectioning (referred to as microtomy), electron microscopy, and finally atom probe tomography on the basis of the two main parameters: the volume of analyzed material and the volume resolution, which are intended to be the size of the smallest image element of the digital reconstruction. As highlighted in Figure 1, serial sectioning with Ga^+^ or Xe^+^ ions covers the range between X-ray tomography and the equivalent technique implemented in the transmission electron microscope. However, FIB tomography has the important limitations of being both destructive to the sample and time-consuming [9]. In addition, the workflow of volume reconstruction involves the acquisition of large image datasets and requires significant computational power and precautions to preserve the fidelity of the reconstruction; indeed, the number of experimental images and the consequent damage imparted to the sample, the drift and instability during acquisition, the artefacts, and the misinterpretation of image details may ultimately affect the results. As described in detail in the following paragraph, one of the critical steps for FIB tomography is the extraction of the significant information from the reconstructed volume; this is referred to as segmentation and is currently evolving from a subjective task basically performed and supervised by the operator to automatic algorithms, which may also involve artificial intelligence to perform. 

Technological progress and the commercial availability of reliable, robust, and user-friendly systems have made FIB tomography a routine technique that complements X-ray and TEM tomography to provide a multiscale analysis of the sample that extends from the millimeter down to the nanoscale.

## 2. FIB-SEM Tomography—From the System to the Data Analysis

### 2.1. Dual Beam System

Figure 2 shows the dual beam FIB-SEM configuration, where the raster of the electron beam is coincident with that of the ion beams. The electromagnetic column controlling the electron gun is vertically oriented, as in a conventional SEM, while the ion beam is tilted at an angle with respect to the electron beam. A modern six-axis motorized stage allows rapid positioning and tilting of the sample, maintaining the visibility of the region of interest (ROI) for both beams.

By tilting the stage by about 50° degrees, depending on the specific system geometry, the ion beam impinges perpendicularly and kinematically removes material from the sample surface after a nucleus–nucleus interaction between the accelerated ions and the target material of the sample [10]. This erosion progresses from the surface and exposes a vertical section of the internal structure, which is readily observable with the electron beam. As shown in Figure 2, the SEM image of the exposed section is geometrically stretched along the *z*-axis due to the sample tilting. For this reason, as described in Section 2.4, the isometric SEM view must be restored for a correct 3D reconstruction. The entire collection of different SEM images from the exposed section forms the dataset, referred to as the image stack, which is the basis of the FIB-SEM tomography.

As introduced in the previous paragraph, one of the main issues of slice-and-view reconstruction is the non-isotropy in the tomogram resolution: the resolution in the milling direction (conventionally referred to as the *z*-axis) is limited by the ability of the FIB to produce parallel and planar sections through the specimen, while in the x-y image plane, the resolution is determined by the physical size of the image pixel and ultimately limited by the diameter of the SEM probe.

Resolution along the *z*-axis is inferior due to a combination of limiting factors: the width and intensity profile of the focused ion beam widens as the beam current increases, and there is source instability and beam drift over the inevitably long acquisition time [6]. The technological improvement of the latest systems and the introduction of drift correction and feedback of the ion emission [11,12] guarantee complete automation of the whole process and stability over the required time.

### 2.2. The Experimental Setup

For a successful 3D reconstruction of an ROI, a preliminary preparation of the specimen is necessary: the thermal and the electrical conductivity of the sample are beneficial to the minimizing of the specimen drift and electrical charging, but at the same time, especially if the study in question involves the analysis of internal porosity or heterogenous materials, we must prevent the FIB milling from creating image artefacts. To improve electrical conductivity, the top surface of the sample can be sputtered with carbon, gold, or Cr or a conductive path can be created between the sample and the sample holder using a double-sided carbon or copper tape suitable for electron microscopy. In addition, it is possible to produce a thick, localized deposition of either Pt, C, or W over the region of interest using induced ion beam deposition (IBID) [13]. This technique employs a small nozzle to inject a gaseous metal–organic precursor close to the ROI. The incoming ion and secondary electrons are generated at the beam position, then they effectively dissociate the precursor molecules, resulting in metal deposition over the desired area, while the volatile part is pumped by the vacuum system. This metal layer, usually Pt or Au, protects the region of interest from inadvertent erosion as well as damage, and ion implantation in the subsurface region improves the local conductivity for non-conductive samples [14] and reduces the so-called “curtain effect”, which is the formation of thickness artefacts parallel to the FIB milling [15]. In addition, to protect the internal porosity of a sample, the specimen can be infiltrated with a resin [16,17], which also provides a better planar section and a good contrast for the pore phase in the image processing (see Section 2.4). This method proved to be very useful for this class of samples, and its use in the analysis of porous materials has now become routine.

The following step is the exposure of the cross-section in front of the region of interest by using an ionic current higher than 2 nA. As shown in Figure 3, two lateral trenches are also pre-cut to avoid shadowing effects from the side walls of the cross-section and material redeposition [14,18]. A reference mark for slice alignment can then be created by milling or IBID just outside the analyzed area by the focused ion beam. Although the FIB software control generally allows the slice-to-slice distance to be selected, the actual thickness may be different from the desired value, affecting the final reconstruction. FIB tomography is a time-consuming technique and drift of the beam stage and sample may occur during serial sectioning. Zekri et al. [19] adopted a method to check the actual thickness of each slice: a triangular shape is milled over the Pt protective layer by the FIB, and, using a simple formula, it helps in aligning the image sequence and checking the actual slice-to-slice distance. Liu also suggests the possibility of performing tomography assisted by the positioning of a conductive nanoprobe close to the ROI, reducing the excess of electrons near the region of interest and improving the quality of the final reconstruction [20]. 

The choice of the probe current and the dwell time for the serial sectioning depends on several factors: the area of the exposed section, the milling rate for the target material under investigation, and the instrumental properties of the FIB system [21,22]. Intense currents produce deep and smooth sections in a short time, but they also have limited accuracy due to their larger diameter and the intensity profile of the ion probe. In addition, material redeposition is so high that it can hide small features of interest [23]. Furthermore, low-current beams achieve a finer focusing for their small probe but can produce artefacts due to incomplete cutting. A detailed review of all the FIB parameters is well described in the work of Kim et al. [24]. Ultimately, the selection of the FIB operating parameters depends on the size of the smallest feature to be reconstructed. Holzer et al. [25] proposed a table containing SEM magnification, voxel dimensions, and dimensions of the data volume for the case of ordinary Portland concrete with different grain sizes. It is evident that tiny features require high SEM magnifications to be analyzed, and this leads to a final reconstruction that is quite limited in final volume. For this reason, a compromise must be made between the resolution of the 3D reconstruction and its final dimensions.

### 2.3. SEM Imaging

The interaction between the electron beam and the target sample produces the two fundamental signals used in SEM imaging: secondary electrons (SEs) and backscattered electrons (BSEs). SEs feature a continuous energy spectrum in the 0–100 eV range [26], representing weakly bound valence electrons or, for the metals, conduction band electrons with a binding energy of 1–15 eV [27]. BSEs are essentially electrons of the primary beam that are deflected at a very high angle and with low energy loss by elastic scattering with the nuclei of the target material [28]. This energy difference allows the two types of electrons to be collected separately and then analyzed for the different information they provide. The BSE signal is highly dependent on the atomic number, giving rise to the so-called the “Z-contrast”. Furthermore, SEs have a more localized signal, although their escape depth strongly depends on the physical characteristics of the sample [29]. In addition, SE imaging is also influenced by the type of detector used. The Everhart–Thornley detector collects both SEs and BSEs, but the position of the detector, which is off-axis with respect to the primary beam, favors sensitivity to sample morphology. On the other hand, the SE detector integrated into the SEM columns (commonly referred to as the in-lens detector) is effective for operation at short specimen column distances and low beam energies [27]. As the stepwise FIB milling of the sample requires the exposure of flat, planar sections, it is expected that the conventional SE morphological imaging will provide images with little or minimum contrast. The purpose of highlighting local variations in sample composition or density makes the choice of BSE imaging the recommended choice for tomography.

In the FIB tomography, two parameters need to be optimized during the SEM acquisition: the acceleration voltage of the primary beam and the contrast of the features of interest in the exposed section of the sample. The first term controls the depth and the radius of the interaction volume, from which the secondary and backscattered electrons are generated. Generally, the higher the beam energy, the larger the interaction volume, although, as mentioned above, there also an influence from the chemical elements of the target material and its density [30]. For this reason, the use of a beam energy of 3–5 keV is recommended; the result is an interaction depth comparable to the thickness of a single slice of the tomography. Furthermore, in this beam energy range, the total emission of the BSEs and SEs exceeds the incoming electrons, creating a favorable electronic balance on the sample surface that eliminates the charging of uncoated insulating materials [27]. Furthermore, the contrast is a more subtle argument, and some enhancements can be operated during the data process through some filters or by setting thresholds in the pixel values. In general, the “Z-contrast” generated by the BSEs is an excellent method of distinguishing between elements of different atomic weights. However, the BSE escape depth is too high for the slice thickness of FIB tomography, and for this reason, a combined signal from the in-lens and ET detectors is often used, exploiting the fraction of BSEs revealed by the ET detector and, at the same time, the high resolution of the in-lens detector [27]. An interesting alternative is the energy-selective backscattered (EsB) detector, which is another in-lens detector with a grid able to select the range of BSEs according to their landing energy. Using a low accelerating voltage, the EsB detector can only detect BSEs coming from a region of the sample surface of the size of the electron probe’s dimensions [31]. 

### 2.4. Data Processing

#### 2.4.1. Stack Alignment

The first step in data processing is the alignment of the collected SEM images. This is a very delicate part of the process as it affects the following segmentation, and subpixel accuracy is required [32]. Some algorithms, such as the StackReg [33] or TurboReg [34] plugins for ImageJ, provide a stable and fast procedure. Arregui-Mena et al. have also proposed an alignment based on a least-square algorithm to improve the precision of the y-position correction [9], while Kelly has proposed the use of a scale-invariant feature transformation algorithm [35,36], where relevant features, such as corners or edges, are detected and linked to analogous points of the following images, to compute a possible spatial transformation [37].

#### 2.4.2. Removal of FIB Artefacts

One of the typical image artefacts produced by FIB milling is the “curtaining effect”, where the surface of the cross-section appears to be covered by some vertical stripes (Figure 4). These artefacts are due to material inhomogeneities, underestimation of the ion beam exposure dose, or variations in crystal orientation [38]. Artefact removal is performed by applying a fast Fourier transform (FFT), adopting the vertical rectangular area as a zero-filter mask centered on the vertical axis of symmetry [39].

Another approach to the problem has been taken by Fitschen et al. [40], who developed an algorithm capable of extracting the clean image from two types of image corruptions: stripe-like features, due to different milling rates, and laminar structures caused by incomplete milling of the material. In addition, as mentioned in Section 2.1, the mismatch between the electron and ion column generates a compression in the pixel length of the y-direction of the collected image. This value must be corrected using the following formula:(1)$$y_{r}=y_{h}/\sin\theta$$
where *y_r_* is the corrected value and *y_h_* is the measured y-value, while *θ* is the tilt angle of the sample holder [9]. Other artefacts can be derived from a local charging, due to non-planar surfaces and pore edges producing extremely bright areas [27]. In addition, pore segmentation suffers from contrast variations due to local topography [35]. 

#### 2.4.3. Image Optimization

To improve the ability to localize and distinguish the features of interest during the segmentation process, the contrast and brightness need to be adjusted directly on the single image. Filters like the Gaussian 3D filter or background equalization, such as the “GradientXTerminator”, are commonly used to reduce noise [41,42], while a sigma filter is quite effective in eliminating residual high frequency noise [37,43]. This filter can be applied to correct for the intensity variations in the images produced by the secondary electrons detected by an Everhart–Thornley detector, the image of which depends on the position of the detector in the electron microscope [44]. Gaboreau et al. [31] proposed a rolling ball algorithm to correct for the non-uniform image background, previously used by Sternberg [45], as the first step of the data processing.

#### 2.4.4. Segmentation

Terao et al. [46] divide the segmentation process into three categories: manual, automatic, and semi-automatic processing. The first can achieve highly accurate results, but it is also operator-dependent and time-consuming as the user has to inspect and process all the images. In contrast, automatic processing separates different features of interest in a single image by adopting a global/local thresholding method to the grayscale of the image. One example is the watershed algorithm, which is largely used in a variety of medical segmentation tasks [47]. It works by considering the pixel of an image as a topographic map consisting of valleys and ridges, corresponding to low and high values on the greyscale, respectively. Applying the watershed in a segmentation means dividing the image into regions by connecting areas with the same pixel value. The k-means or the ISODATA algorithm is also widely used in pore segmentation [48]. This method defines the peaks of the histogram of the image and distributes the data points so that the sum of the squared distance to the center of the cluster is minimized [49,50]. Another method used in FIB tomography is the Otsu threshold, which performs automatic image thresholding by minimizing the intra-class intensity variance [51]. An example of different segmentations of the same image and their influence on the porosity calculation is shown in Figure 5.

Another automatic thresholding method is the one developed by Niblack [52] in 1986 and later implemented by Savuola and Pietikainen [53], where the image binarization is obtained by performing several local thresholds using formulae that consider the mean and standard deviation of the surrounding pixels. However, a fully automatic segmentation suffers in terms of accuracy because it is based only on the brightness of the image, where the presence of some local charging can lead to a misinterpretation of the sectioning. Furthermore, a semi-automatic approach was developed by Thiele et al. [54]; this approach optimizes the threshold values in a first step, followed by a second manual checking to improve the result. Salzer et al. proposed a similar analysis called threshold propagation, where they compare grey values in the z-direction to evaluate the appearance and disappearance of structures [55,56]. In the work of Kelly et al., two consecutive thresholding segmentations are applied, the second of which, named “soft thresholding”, is specific to pore recognition [35]. Goral et al. [57] apply a machine learning approach using the Zeiss Zen Intellesis software, which adopts the “Forest of Randomized trees” method to build a multivariant image segmentation model on the generated features [58,59,60].

## 3. FIB-SEM Tomography—Application in Materials Sciences

Since the first reconstruction operated by Wilson et al. in 2006 [61], FIB tomography has been applied to a wide range of research fields, from Li-ion batteries to the analysis of the shale rocks, often providing interesting multiscale analyses by its combination with X-ray microscopy (XRM) and TEM tomography, improving the spatial resolution from the micron to the nanoscale. Moreover, these morphological data can also be integrated with additional information on the crystalline structure or elemental distribution from other in situ techniques or spectroscopy, such as electron backscatter diffraction (EBSD) or EDX, further increasing the importance of this type of tomography. In the following subsections, a detailed view of the different applications of FIB tomography is given, with all the articles in which this technique is used, classified by research topic, and a listing of all the possible information that can be extracted from its application.

### 3.1. Fuel Cells

The first significant application of the FIB-SEM tomography was the reconstruction of a Ni-Y-stabilized zirconia (Ni-YSZ) composite anode for solid oxide fuel cells (SOFC), as shown in Figure 6, where the authors connected the microstructure to the electrochemical performance of the cell by extrapolating parameters such as porosity, three-phase boundary (TPB) length, or tortuosity [61]. However, the identification of the porosities during the segmentation process resulted in being quite critical. In addition, Shearing et al. found an over-assignment of the YSZ phase during the segmentation, with respect to the Ni and pores, due to the overlapping with the greyscale histogram of the Ni phase, generating a significant error in the calculation of the TPB value [62]. The work of Iwai et al. [63] defined the first procedure to calculate TPB values using the centroid method, where a TPB is formed by a triangle made of three voxels containing three different phases (Ni, Y, and pores). The calculation of this parameter is given by the distance between the centroids of these neighboring triangles. These authors were also the first to introduce the practice of infiltrating these porous samples with a resin. This procedure was resumed in 2011 by Joos et al. [17], who treated an SOFC cathode with a two-component resin to improve the planar sectioning and the contrast between the pores and the electrode material. They also resolved the segmentation by applying the Otsu method [51] for the first time to calculate the best value for the threshold. Vivet et al. [64] developed an improved algorithm for the calculation of the TBL in Ni-YSZ cermet anodes for SOFC, while the paper of Cronin et al. combined electrochemical impedance analysis with the porosity obtained by FIB-SEM tomography and showed a strict correlation between the pore evolution and the polarization resistance of Ni-YSZ fuel cell anodes [65]. The effect of sintering on the fabrication of these electrodes has been investigated by Holzer [66], using the continuous phase size distribution (c-PSD) elaborated by Münch [67] for the pore size distribution, while, in the work of Song [68], microstructural changes related to the redox cycles at high temperature are associated with deterioration of the mechanical properties and polarization resistance. In the paper by Trini [69], phase-field simulations based on microstructural parameters derived from FIB-SEM tomography were carried out on a Ni-YSZ cell made up of 25 stacks and run for 9000 h to observe and understand the degradation mechanisms of a prolonged operation.

A first attempt to combine X-ray computed tomography (XCT) with the FIB-SEM tomography was made by Wargo, to study of the contributions of the gas diffusion layer (GDL) and the microporous layer (MPL) regions on the transport by the diffusion media [50]. Later, a similar experiment was also proposed by Göbel et al. for evaluating two different types of GDL materials [70] (Figure 7).

Porous ceramic films of La_0.6_Sr_0.4_Co_0.2_Fe_0.8_O_3−δ_, used as cathodes in SOFCs, have been extensively studied in the works of Chen [71,72] and Endler-Schuck [73], while Almar and co-authors have extended their investigations on the oxygen transport kinetics of Ba_0.5_Sr_0.5_Co_0.8_Fe_0.2_O_3−δ_ (BSCF) cathodes [74]. Mechanisms in anodic and cathodic polarizations for screen-printed La_0.6_Sr_0.4_Co_0.2_Fe_0.8_O_3−δ_ with a thin interlayer of gadolinium-doped ceria have been studied using a microstructure analysis of 14.9 × 14.9 × 14.9 μm^3^ realized with FIB-SEM tomography [75]. The same type of electrode was also characterized by Kishimoto et al. [76], who created a 1D numerical model to investigate the effect of the microstructure on the final performance, starting from a FIB-SEM tomography with 12.5 nm^3^ voxel size, by extrapolating parameters such as the tortuosity factor, surface area, and distribution of the pores and particles. Rhazaoui et al. applied FIB-SEM tomography as a geometrical input into the ResNet model, which was used to calculate the effective electronic and ionic conductivities of a Ni/10ScZ anode [77,78]. In this method, the 3D microstructure is represented in voxels and a potential difference is applied to this network, where, by Kirchhoff’s law of current conservation, it is possible to extract the equivalent resistance and conductivity of the whole structure. In the work of Yan et al. [79], the structure of La_0.6_Sr_0.4_Co_0.2_Fe_0.8_O_3−δ_ powders is reconstructed by FIB-SEM tomography and used as a starting model for a discrete element approach aimed at predicting the effects of the sintering process. A correlative tomography technique for this class of materials has been developed by Wankmüller and co-workers [80] to visualize the spatial organization of primary and secondary phases at the interface of La_0.6_Sr_0.4_Co_0.2_Fe_0.8_O_3−δ_/10 mol% gadolinia-doped ceria/8% mol% yttria-stabilized zirconia electrolyte. This method correlates the typical 3D FIB-SEM reconstruction with the elemental distribution of the interface obtained by EDX spectroscopy in a transmission electron microscope operated in scanning mode (STEM) (Figure 8).

In addition, the influence of different manufacturing methods on the microstructure of the same class of electrodes has been shown by Singh et al. [81], while the work of Zekri uses FIB-SEM tomography to evaluate the microstructure degradation of this type of anode after a long period of operation [19]. La_0.6_Sr_0.4_Co_0.2_Fe_0.8_O_3−δ_ symmetric electrode cells with Gd_0.1_Ce_0.9_O_1.95_ (GDC) electrolytes and their corrosion mechanisms have been extensively studied in the papers of Wang [82,83,84] and Miyahara [85]. Anisotropies in the microstructure of La_0.6_Sr_0.4_Co_0.2_Fe_0.8_O_3−δ_ film deposited on rigid GDC have been investigated by Yan et al. [86], demonstrating a preferred pore orientation and elongation that increase with sintering time or temperature. Microstructural parameters, such as phase volume fraction, grain morphology, contiguity of the phases, and TBL, have also been evaluated for nickel-samaria-doped ceria (Ni-SDC) synthetized by the citrate–nitrate combustion reaction [87]. The morphology and porosity of platinum group metal-free (PGM-free) iron-nitrogen-carbon (Fe-N-C) catalyst layers for the oxygen reduction reaction were investigated by Stariha et al. [88] The LaPrNiO_4+δ_ electrode for intermediate temperature solid oxide cells (IT-SOC) and its durability have been extensively studied by Khamidy and co-authors [89] by combining electrochemical data with microstructure information obtained by FIB-SEM tomography and X-ray diffraction and fluorescence. La_0.5−x_Pr_x_Ba_0.5_CoO_3−δ_ cathodes for low temperature solid oxide fuel cells (LT-SOFC) have been characterized in the work of Garces [90], with information on the O_2_-reduction mechanism and its kinetic coefficients, O-ion diffusion and O-surface exchange, from the microstructure data generated by FIB-SEM tomography.

The first 3D reconstruction of a self-humidifying membrane electrode assembly for proton exchange membrane fuel cells (PEMFC) was performed by Jung in 2016 [91]. A 1 μm^3^ volume was reconstructed by the manual segmentation of a stack of 67 images, with a 15 nm spacing and SEM pixel resolution of 1.5 nm × 1.5 nm. Okumura investigated the cathode microstructure with a different amount of Nafion^®^ ionomer, using TEM imaging to highlight the distribution among Nafion^®^, carbon support, and platinum nanoparticles [92], while Vierrath and co-workers suggested an alternative method to enhance the image contrast of the catalyst layer by filling the matrix pores with ZnO deposited by atomic layer deposition [93]. The collapse of the porous cathode microstructure was characterized in the work of Star et al. [94] by the correlation of electrochemical methods, infrared spectroscopy, and FIB-SEM tomography, showing that the platinum ripening and carbon black corrosion were the main causes of the performance loss. Grunewald and co-authors presented a lattice Boltzmann method model applied to a cathode catalyst layer (CCL), which combines tomography data and mesoscale modeling techniques, to improve the knowledge of the transport mechanisms of the oxygen in the catalyst layer [95]. In an article by Pournemat, a voxel-based Montecarlo model describes the strict relationship between the wettability and pore size distribution on the water distribution within the gas diffusion layer (GDL) and the CCL [96], while Nakajima used FIB-SEM tomography to model the pore network of their hydrophobic microporous layers to evaluate the convective air permeability and oxygen diffusivity [97]. A similar work has been published by Maloum et al., who used the data collected from X-ray and FIB tomography to construct an innovative numerical computation for the evaluation of the microporous layer in a fibrous GDL [98]. Furthermore, new membrane technologies with low environmental impact are being developed using a flow-processing technique in which the Nafion matrix is filled with aligned zeolite nanosheets fillers, and the suitable orientation for the zeolite nanosheet is characterized by the FIB-SEM sectioning [99]. The microstructure of the catalyst layers bonded by polybenzimidazole for high temperature fuel cell have been investigated by Prokop et al., who focus on the distribution of platinum in the catalyst layers and its degree of penetration into the electrode [100].

FIB-SEM tomography has also been applied to direct methanol fuel cells (DMFC) to investigate the ageing effects on the microstructure of the anode catalyst layer (ACL) after complete methanol starvation [44]. The sample was embedded in epoxy resin, polished, and then sputter-coated with a layer of gold. In addition, micrometer resolution synchrotron X-ray tomography was performed on the entire membrane electrode assembly (MEA) to obtain a complete view of the system.

### 3.2. Batteries

The first paper reporting an application of the FIB-SEM tomography to the Li-ion batteries was published in 2011 by Ender et al. [101]. A simple cathode made of LiFePO_4_ commercial powders, mixed with carbon black and polyvinylidene fluoride (PVDF), was characterized by electrochemical impedance and the reconstruction of a 5 × 5 × 15 μm^3^ volume from a stack of 200 SEM images (Figure 9). They also extracted volume fractions, volume-specific surface areas, and tortuosity for the three individual phases. This work was also very important for the introduction of a silicon resin as an embedding material, which was able to optimize the image contrast for the pores against the carbon black and LiFePO_4_. The method was improved the following year by the adoption of an advanced local threshold method that took into account gradients in the luminosity of the neighboring voxels [102].

A similar approach was used by Eswara-Morthy, where the contrast of the porous carbon electrodes was enhanced in situ by adding a thick layer of Pt (12 μm) by ionic beam-induced deposition (IBID) [103]. In their 2015 article, Wieser and co-authors [104] realized a multiscale reconstruction of a simple lithium-ion battery electrode to understand the role of the μm scale graphite, which was acting as an active material, and the nm scale polymeric binder.

Synchrotron radiation computed tomography (SR-CT) was used to detect the active materials, while FIB tomography was used to distinguish the polymeric binder from the pores, achieving a voxel size of 5 nm × 6.27 nm × 10 nm. This work also applied the two-step segmentation method developed by Prill [32]. A similar multiscale approach has also been used to model a LiCoO_2_ cathode, incorporating micro- and nanoscale information for an improved calculation of the 3D transport properties [105] (Figure 10) or to characterize the morphology and charge transport limitations of LiNi_1/3_Mn_1/3_Co_1/3_O_2_ (NMC), LiFePO_4_ (LFP), and blended NMC/LFP electrodes for electric vehicle batteries [106,107]. Another investigation of the same type of electrode has been carried out by Cadiou and co-workers, who used the combined microstructural information from the X-ray and FIB-SEM tomography to perform an electrostatic simulation using the fast Fourier transform (FFT) method, obtaining a good numerical fit with broadband dielectric spectroscopy for the bulk conductivities of the C/LiFePO_4_ and the carbon black/poly(vinylidene fluoride) phase [108]. Vierrath studied in detail the distribution of the carbon binder domain (CBD) for this class of batteries and found a correlation between the inhomogeneity of the CBD and a reduction in the electron conductivity [109]. Furthermore, in a recent paper, Almar’s group conducted a study of the microstructural features of the positive electrode for four commercial batteries and two high-power and two high-energy Li-ion battery consumer cells [110]. Using appropriate 3D analysis techniques, they obtained quantitative parameters to characterize the active material, the carbon black binder, and the pore phases. A different approach to improving the contrast between the three phases, the LiCoO_2_ particles, the carbon-based materials, and the electrolyte has been proposed by Liu and co-authors, who infiltrated the sample with a silicon resin, as suggested by Ender [101], and produced a very smooth cross-section with a triple ion beam cutter operating at a 4 kV accelerating voltage and 2 mA ion current for 8 h [111]; Biton and co-authors have suggested an infiltration with a brominated (Br) epoxy capable of producing an enhanced contrast with respect to the C- and Si-based resins [112].

An investigation related to the structural changes for long-term degradation has been addressed in the work of Song, revealing an evolution in the damage of the active material [113], while Scipioni and co-authors have demonstrated the presence of amorphous carbon surrounding the LiFeO_4_ in the degraded electrode [114]. Similarly, Etiemble and co-authors [115] analyzed the evolution of the 3D microstructure of a silicon/carbon/carboxymethylcellulose electrode for Li-ion batteries before and after 1, 10, and 100 charging/discharging cycles. The evolution of morphological features, such as volume fraction, spatial distribution, size, connectivity, and tortuosity proved that the major changes in the electrode are due to a variation in the size and shape of the Si particles and to the cracking of the electrode, which leads to a solid electrolyte interphase. A characterization of a commercial 2.5 Ah LiFePO_4_/graphite 26,550 cylindrical cell, consisting of XRD, XPS, and FIB tomography, was carried out by Scipioni et al. They observed the presence of microsized carbon-based agglomerates, probably due to the electrolyte decomposition [116]. Furthermore, pristine and cycled LiNi_x_Mn_y_Co_1−x−y_O_2_ (NMC) and Li(Li_0.2_Ni_0.13_Mn_0.54_Co_0.13_)O_2_ have also been analyzed in their microstructure parameters, showing that, for both electrodes, the cycling creates a continuous detachment of the carbon-doped binder from the active particles [117]. Another significant example of correlative tomography, FIB-SEM, and synchrotron X-ray tomography, performed in the same specific spot, was carried out by the Moroni group on a lithium manganese oxide composite cathode, validating the segmentation technique for X-ray tomography on a one-hundred-micron scale reconstruction [118].

Metal–air batteries have been characterized by FIB-SEM tomography, starting from the work of Danner [119], which analyzed the efficiency of porous silver as a catalyst layer; in addition, the works of Biton and Yufit reconstructed the evolution and dissolution of the Zn dendrites, which are probably one of the main causes of degradation in Zn–air batteries [120,121]. Other works worth mentioning with regard to this application include Prill’s 2017 paper, which demonstrated the dependence of the transport and electrical properties on porosity distribution for a carbon electrode [122], while Lagadec and co-authors reconstructed the microstructure of polyethylene and polypropylene separators to simulate their deformation under compressive strain [123]. FIB-SEM tomography has also been applied to understand the high performance of a silicon–graphene hybrid material used as a negative electron in lithium-ion batteries [124].

All the articles cited in this section are summarized in Table 1, where the main features of the applications of FIB-SEM tomography in batteries are presented.

### 3.3. Solar Cells

Despite the great importance and number of articles published annually, there are only two papers to report here that use FIB-SEM tomography, and they are listed in Table 2. The first is by Wollschläger [125], who characterized the role of the porous TiO_2_ nanoparticle film in dye-sensitive solar cells (DSSC), combining the morphological 3D FIB-SEM reconstruction with the structural information obtained by the transmission Kikuchi diffraction (TKD) for electron transparent samples, to understand the role of the crystal orientation of the grains inside the 3D pore network. The other work is from Suter’s group, which characterized two different types of photo-electrodes: the 650 nm thick hematite (α-Fe_2_O_3_), deposited on FTO by atmospheric pressure chemical vapor deposition (APCVD), and the 7.5 µm thick lanthanum titanium oxynitride (LaTiO_2_N), fabricated by electrophoretic deposition on FTO substrate, linking the specific morphological information obtained from the FIB-SEM tomography to the multi-physical transport characterization [126]. According to the authors, this characterization method is very promising due to its capability to control and enhance the performance of the photoelectrodes. Another example of FIB-SEM reconstruction is that from Andrzejczuk [127], who compares the morphology of titania nanotubes, obtained by electrochemical anodization [128], with electron tomography. Although the latter method is more accurate due to its high resolution, FIB-SEM tomography analyzes a larger part of the sample, making its measurement less dependent on the irregularity of the sample.

### 3.4. Ceramics

Although the first definition of FIB-SEM tomography can be traced back to the paper of Inkson [129], the first work using the current dual beam configuration, was carried out by Holzer and co-workers in 2004 [14]. Their 3D reconstruction of a BaTiO_3_ electroceramic in its highly porous state (Figure 11) was a turning point in this type of analysis as they also established a definite workflow for the data processing of the pore analysis, which was later refined in their 2007 article [67].

Two years later, the same authors applied this technique to samples of ordinary Portland concrete with different grain sizes in order to perform statistical shape particle analysis and a topological characterization [25]. This work also implemented the use of reference marks as part of an automated drift correction procedure. The work of Schaffer et al. [130] introduced the possibility of combining the morphological reconstruction of the FIB-SEM analysis with the elemental analysis given by the energy-dispersive X-ray spectroscopy (EDX). In this way, it was possible to distinguish not only the pore network of the sample but even the spatial distribution of the CaTiO_3_ and Mg_2_TiO_4_ within the ceramic matrix (Figure 12). Jiang mixed information from X-ray and FIB-SEM tomography to analyze the pore characteristics, including porosity, pore size distribution, pore shape, orientation, connectivity, and tortuosity, of cement paste made of ordinary Portland cement [131]. FIB-SEM tomography has also been used to characterize the damage corrosion of the hardmetals, such as the WC-Co cemented carbide [132]. The evolution of the cermet during the sintering process has been studied in the work of Pötschke et al. [133], while a multiscale approach for the study of the porosity of α-Fe_2_O_3_, obtained by spark plasma sintering [134], has been proposed by Papynov [135].

### 3.5. Metal, Steel, and Alloys

The first article to be reported in this section is that by Maleki et al. [136]. From the analysis of the morphology at different scales, obtained by combining synchrotron X-ray and focus ion beam tomography, they investigated the relationship between the microstructure evolution and the deformation behavior of Sn-4.0Ag-0.5Cu solder during isothermal ageing. The Yan group investigated the impact of grain boundary carbon brittle on the microstructure of a biomedical Ti-15Mo alloy and how it affected the fatigue properties and the corrosion resistance [137]. The task was accomplished by identifying these carbon-enriched phases as face-centered cubic Ti_2_C by TEM analysis and by showing a distribution like the primary α-Ti by FIB-SEM tomography. The distribution and composition of the γ′ and γ″ phases of the Inconel 718, a corrosion-resistant nickel-based superalloy, have been reconstructed Kulawik’s group using FIB-SEM tomography and STEM-EDX elemental maps [138]. Kruk wrote some articles on Allvac 718Plus, a Ni-based superalloy with improved performance compared to the Inconel 718 [139,140,141]. He and his group combined STEM-EDX, electron, and FIB-SEM tomography to investigate the microstructural features, composition, and distribution of the different phases. The effects of the underwater wet welding, performed on rutile-coated electrodes with a low carbon steel wire core, were characterized by Silva through a multiscale approach, involving X-ray micro-CT, synchrotron micro-CT, and FIB-SEM tomography, to study pores, cracks, and inclusions [37]. Roland and co-authors developed a model to estimate the mechanical stress–strain curve for the strontium-modified Al-Si alloy from the real 3D coral-like morphology of the eutectic Si in the Al–Si alloy [142], and another similar research study has been conducted on the Al7075 alloy by the Singh group [143]. Micro-CT and FIB-SEM tomography have been used to characterize the secondary phases of the eutectic phase mixture, (α-Mg + MgZn) and (α-Mg + Ca_2_ + Mg_6_ + Zn_3_), of the as-cast Mg-3Zn and Mg-3Zn-0.3 Ca alloys [144]. The presence and distribution of the Fe-rich a-Al_14_Fe_3_Si_2_ with a bcc structure and the τ2 phase Al_4.5_FeSi were observed by FIB tomography in unmodified and Sr-modified Al-10Si-0.3Fe casting alloys, and the different phases were confirmed by electron diffraction with the TEM [145]. Micro-bending beams of the fcc nickel-based superalloy CMSX-4 were first machined by the FIB, then subjected to the fatigue experiment, and finally, after 6100 load cycles, its microstructure was reconstructed by FIB-tomography [146] (Figure 13). The μ and P phase precipitates of this alloy have also been determined by a combination of different techniques including SEM, TEM, high-angle annular dark field (HAADF), FIB tomography, and selected area diffraction with beam precession (PED), EDX, and energy loss spectroscopy (EELS) [147]. Yang et al. combined correlative electron microscopy, FIB-SEM tomography, and atomistic simulations to explain a corrosion phenomenon called 1D wormhole corrosion, which is responsible for the extremely rapid infiltration of a molten fluoride salt into a Ni-20Cr alloy [148].

The literature shows many studies dedicated to different types of steels, in which FIB-SEM tomography has been successfully applied. Wongpromat [149] focused his research on the early stages of the oxidation of AISI 441 at 800 °C and under 5% H_2_O in O_2_, to explore its application as a metallic interconnection within the SOFC. Burnett and co-authors [150] examined a sample of AISI 316 from power station steam to investigate the reheat cracking through the combined use of X-ray CT, FIB-SEM tomography, EDXS, and TEM. Kawano studied the dispersion of intergranular NbC precipitates in Nb-added austenitic stainless steel, associating the dislocations with the presence of these precipitates in the {111} slip plane [151]. Maetz investigated the effect of ageing in 2101 lean duplex stainless steel [152]. The correlation between grain boundary characteristics, morphology, and dispersion of intergranular carbides in 347 austenitic steel was obtained in the work of Ochi [153], who identified three different types of carbides: two planar types, with an orientation relationship with one of the neighboring grains, and a rod type, probably associated with the defects of the steel. The same author also described the intra- and inter-granular precipitates, consisting mainly of Cr_2_N and Cr_3_Ni_2_Si(N), for the nitrogen-added austenitic stainless steel SUSXM15J1 [154]. The growth of a single graphite nodule in a near-eutectic ductile iron was reconstructed using FIB tomography in the work of Ghassemali et al. [155]. The effect of long-term oxidation (25,000 h) at 700 °C on the austenitic steel Sanicro 25, commonly used in heaters and superheaters, was evaluated by the Cempura group [156]. Their research shows changes in the microstructure of the steel at a thickness of about 20 μm due to the formation of a protective Cr_2_O_3_ oxide and the consequent precipitation of Fe_2_W. They also confirm that the steam oxidation affects the bulk steel, coarsening the carbide particles at the grain boundaries. Recently, Photer-Simon has published a paper on the corrosion effects on stainless steel 304L due to a KCl-rich environment at 600 °C, using FIB-SEM tomography and TEM to understand the grain boundary attack [157].

The synthesis of porous metal microstructures based on an Sb–Cu alloy and obtained by the additive expansion by the reduction of oxides (AERO) process has been characterized in the work of Atwater [158], while Hu et al. have prepared and characterized porous Cu-Cr and Ag-Ni composites by a crucible-free liquid metal dealloying process using FIB-SEM tomography [159]. The porosity of nano-silver joints for applications in SiC technology has been studied by Rmili et al. [160], while the effect of thermal etching in an oxygen-rich atmosphere on the catalytic activity of silver has been addressed in the work of Ollivier [161]. The growth mechanism of primary Cu_6_Sn_5_, intermetallics considered promising as primary crystals in bulk solder [162], were studied in Sn–Cu alloys and solder joints by combining EBSD, FIB tomography, and synchrotron radiography [163]. Synthesis by chemical vapor deposition, such as 1D Al/Al_2_O_3_ nanostructures or SnO_2_ nanowire, have been extensively studied by coupling 3D FIB-SEM reconstruction and atomic force microscopy (AFM) [164,165]. A similar study was carried out by Mangipudi, who analyzed the structure of gold nanoparticles with ligament sizes on the order of ten nanometers [166]. A more comprehensive work on the same type of material was made by the Jeon group [167], which used FIB reconstruction data to recover the distribution of ligament size, surface-to-volume ratio, and scaled connectivity density for samples with different ligament sizes prepared by heat treatment. Van der Hoeven et al. were able to obtain not only the size distribution of an assembly of gold-coated silica nanorods but also the orientation of a single particle [168]. The microstructure and the mechanical behavior of Al-SiC nanolaminates, obtained by magneton sputtering deposition, were the focus of the work by Mayer et al. [169]. The response to nanoindentation for a silver-based nanoporous material, obtained by a selective chemical etching of an Ag_38.75_Cu_38.75_Si_22.5_ crystalline alloy, was determined in the work of Champion [170]. The 3D FIB reconstruction was used to understand the mechanical behavior during this type of analysis.

### 3.6. Geology

Since the work of Sondergeld in 2010 [171], where FIB-SEM tomography was performed as a technique to study the porosity of shale rock, this type of analysis has become quite common in this field of research. A precise and detailed knowledge of the real porosity of a shale rock becomes important data with which to extract information about the mechanisms that regulate the enrichment or accumulation of the gas inside the rock. Often supported by other investigation methodologies, directly (BET or mercury porosimetry) or indirectly (TEM tomography or X-ray micro-CT), the 3D FIB reconstruction has become increasingly important over the last five years, even reaching a voxel size of a few cubic nanometers. In addition, the growing interest in so-called correlative microscopy has led to the creation of workflows that bridge the gap between the nanometer scale and macroscopic observation, integrating different analysis techniques in a single result. An example can be found in the work of Ma [172], where a multiscale approach, integrating different methodologies, has been proposed (Figure 14). A list of papers dedicated to the shale rock where FIB-SEM tomography is performed is presented in Table 3.

Regarding the earth sciences, other works propose the reconstruction of microbialites [189], clays [18,190,191,192], zeolites [193,194], Fe-rich olivine [195], dolomites [48,196], coals [197], soils [198], and even samples of urban dust [199]. In addition, Zhou et al. investigated the formation of Au nanoparticles in porous low-Si magnetite by analyzing nanoscale structure and crystallography [200].

### 3.7. Materials for Nuclear Energy

This paragraph summarizes the papers that have applied FIB-SEM tomography in the nuclear energy field. For example, Keller applied this technique to MX80 bentonite samples to study the evolution of the intergranular pores under conditions similar to those found in nuclear waste repositories [201], while Hemes used a combination of micro-CT, BIB-SEM, and FIB-SEM tomography to reconstruct the Oligocene age Boom clay, which is considered to be a potential host material for radioactive waste disposal in Belgium [202]. Bulk plutonium and uranium, as well as the distribution of plutonium oxide particles in the plutonium oxalate precipitates and UO_2_ bubbles produced in high burnup, were also analyzed [203,204,205]. Baris compared the 3D microstructure of Ziraloy-2 LK3/L, used as a cladding material in a Swiss reactor, under the conditions of high and low irradiation from a boiling water reactor (BWR) [206,207]. Another important paper is that of Arregui-Mena and co-authors on the porosity of an AGX graphite, used as a moderator for fast neutrons, and the effect of irradiation on its microstructure [9,208].

### 3.8. Fibers and Polymers

SEM observation of carbon-based materials, such as fibers or polymers, is always a delicate task because they are easily degraded under the high vacuum conditions of the microscope. In addition, their morphology can be quickly altered by the milling action of gallium ions, even with very short exposure times. For this reason, specific treatments are required to preserve their structure, as well as the presence of heavy elements to enhance image contrast. For example, to study the internal network of the nanofibers obtained by electrospinning of Polyamide 6 (PA6), the Stachewiczadopteda protocol, in which the sample is first wetted in a solution of aqueous iodine and then flash-frozen under liquid nitrogen, is followed [209]. This procedure fills the internal voids of the fibers with an amorphous solid derived from the frozen iodine solution, thus obtaining a better polished cross-section with the focused ion beam. In another paper, the same author analyzed the wetting mechanism of these nanofibers with a low-surface-tension oil by using the cryo-FIBSEM in combination with an atomic force microscope (AFM) for the contact angle measurement of a single fiber [210]. A similar protocol has also been used to understand the interaction between osteoblasts and PLGA fibers [211,212]. The Campo group were successful in the alignment and the phase separation analysis of a polydimethylsiloxane/poly(methyl methacrylate) PMMA/multiwall carbon nanotube (MWCNT) electrospun composite [213], resolving the contrast mechanism between the polymer and the MWCNT by synchrotron spectroscopy and helium microscopy. Diblìkovà and co-authors published a paper in which they reconstructed three mixed-matrix membranes of polymide-silicalite [214]. Another significant work relates to highly porous particles for protein sorption, made from the polystyrene-b-poly (acrylic acid) (PS-b-AA) copolymer, where the authors made an interesting comparison between two different types of tomographic techniques, the FIB-SEM, and the serial block face (SBS) technique, in which the sample is cut by an ultramicrotome mounted inside the SEM chamber and each pristine section is imaged using BSEs (Figure 15) [215].

The nanostructured network of molecularly imprinted polymers (MIP) has been obtained by Neusser [216], while Aslannejad et al. have characterized the hydraulic properties of the paper coating layer from the total porosity and permeability values extracted by FIB-SEM tomography [217]; they also created a model for the water imbibition for coated and uncoated papers [218]. The work of Roberge et al. [219] analyzed the structural characteristics of commercially available membranes for microfiltration and ultrafiltration made of polyacrylonitrile (PAN) and polyethersulfone (PES), respectively, using specific staining agents to increase the image contrast of the fibers.

## 4. Conclusions

As shown in the article, FIB-SEM tomography has proven to be a mature and reliable technique, whose applicability to different materials and research fields has been demonstrated; it is a technique which can be rapidly integrated with other analytic techniques. FIB-SEM tomography plays a crucial role in the development of innovative multiscale and multimodal correlative microscopy workflows as it can be seamlessly integrated with other imaging modalities. As shown in the introduction and some applications, FIB-SEM tomography can fill the gap between the non-destructive X-ray families of tomographic techniques, which provide sub-micron resolution, and the nano- to atomic-scale resolution achieved by TEM tomography. As correlative microscopy is defined as a specialized approach in scientific imaging and analysis that involves combining multiple characterization techniques to obtain a comprehensive understanding of the specimen, future technological developments are expected in terms of automation in the detection of the specific ROIs by different instruments and in the creation of particular sample holders that can facilitate the transfer from instrument to instrument. Another important improvement will come from the integration of the advanced machine learning and deep learning systems that are to be employed in data acquisition, analysis, and segmentation algorithms, enabling more efficient and accurate extraction of information from complex 3D image datasets, automatically removing noise and artefacts of the images. In addition, although the technological progress has improved the automatic control of the focused ion beam, one of the negative aspects of this technique remains its duration, which can even reach a whole day for the cutting and imaging of large volumes. A solution can be found in the adoption of the latest plasma FIB models, which use xenon ions instead of gallium, promising a significant reduction in the process time, while maintaining a nanometric resolution. Another negative aspect of this technique can be seen in the small number of articles on carbon-based materials. Undoubtedly, light materials are not efficiently cut by the gallium ion beam without a correct, complex, and time-consuming sample preparation, but a lot of procedures have been created in the life sciences world, as demonstrated by the long list of scientific papers available in the literature that use this reconstruction technique. However, the recent explosion of cryo-microscopy is leading to a gradual diffusion of these instruments, in particular the cryo-FIB-SEM, which also guarantees a promising future for the analysis of light materials, allowing a complete three-dimensional reconstruction with a very limited sample preparation.

## Figures and Tables

**Figure 1 materials-16-05808-f001:**
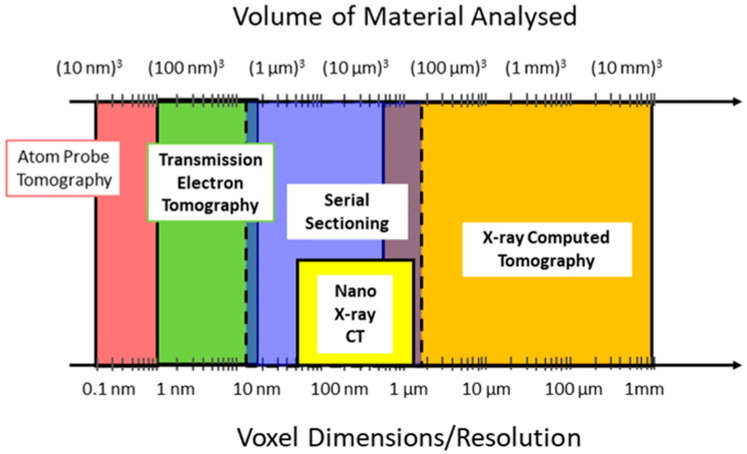
Comparison between different 3D tomography techniques for volume of analyzed material and voxel resolution. The dashed line is for soft materials.

**Figure 2 materials-16-05808-f002:**
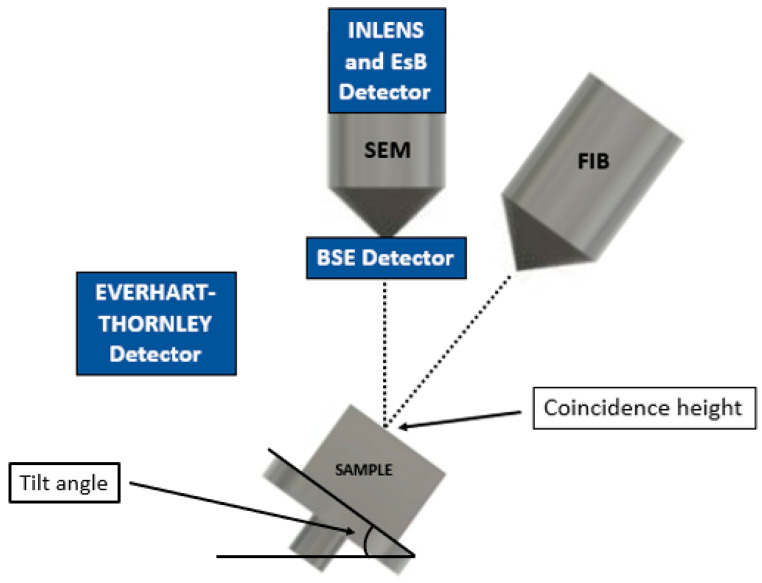
Schematic configuration of sample, columns, and detectors inside the dual beam FIB-SEM system, with the specimen region of interest (ROI) at the coincidence height, where both the electron and ion beam converge at the same position on the sample surface and at a specific tilt angle. The position of the different detectors inside the instrument is also shown, with the Everhart–Thornley and BSE detectors positioned laterally in the sample chamber and just above the end of the electron column, respectively.

**Figure 3 materials-16-05808-f003:**
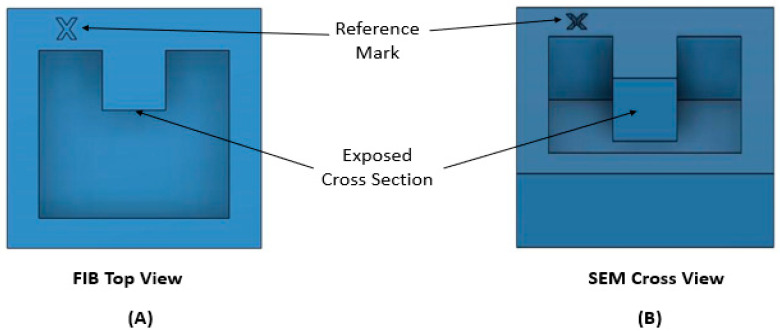
Scheme of a typical setup for the FIB tomography, showing different FIB and SEM points of view of the exposed cross-section: (**A**) FIB top view; (**B**) SEM tilted view of the cross section.

**Figure 4 materials-16-05808-f004:**
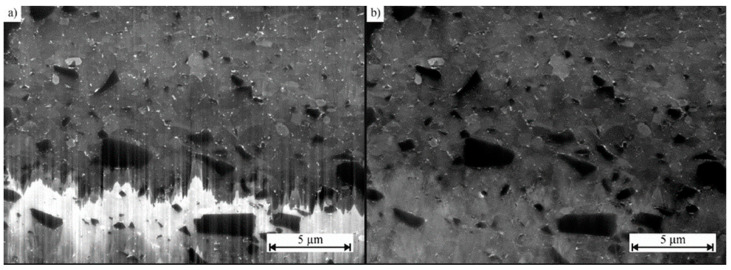
FIB cross-section of an aluminum matrix filled with SiC particles; (**a**) shows the original image, in which the curtain effect is evident, while (**b**) is the FFT-filtered image [38].

**Figure 5 materials-16-05808-f005:**
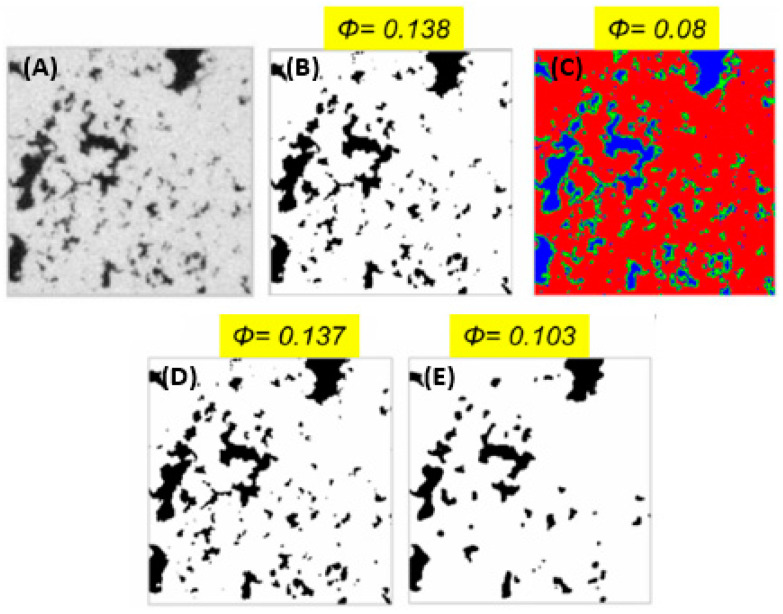
Different types of segmentation for the calculation of the porosity (φ) on Silurian dolomite image: original 2D greyscale image (**A**), Otsu (**B**), k-means (**C**), manual (**D**), and watershed segmentation (**E**) [48].

**Figure 6 materials-16-05808-f006:**
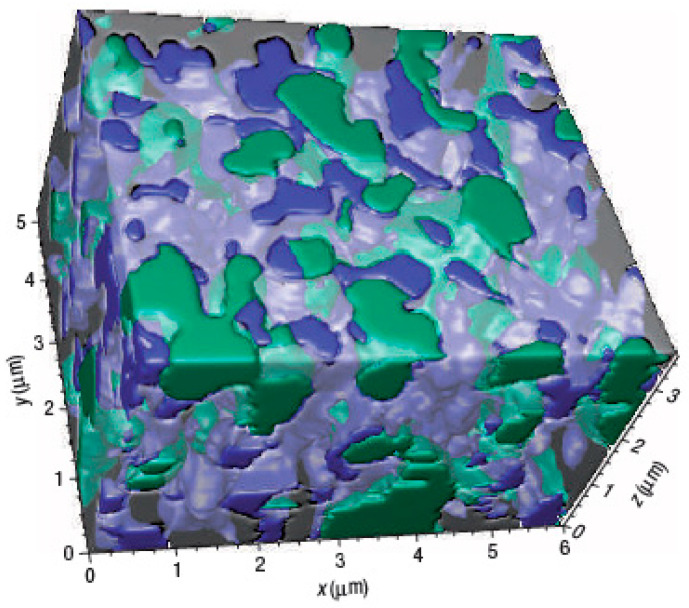
The first 3D FIB reconstruction of Ni-YSZ anode for SOFC taken from the work of Wilson et al., where the three phases are Ni (green), YSZ (gray), and pores (blue) [61].

**Figure 7 materials-16-05808-f007:**
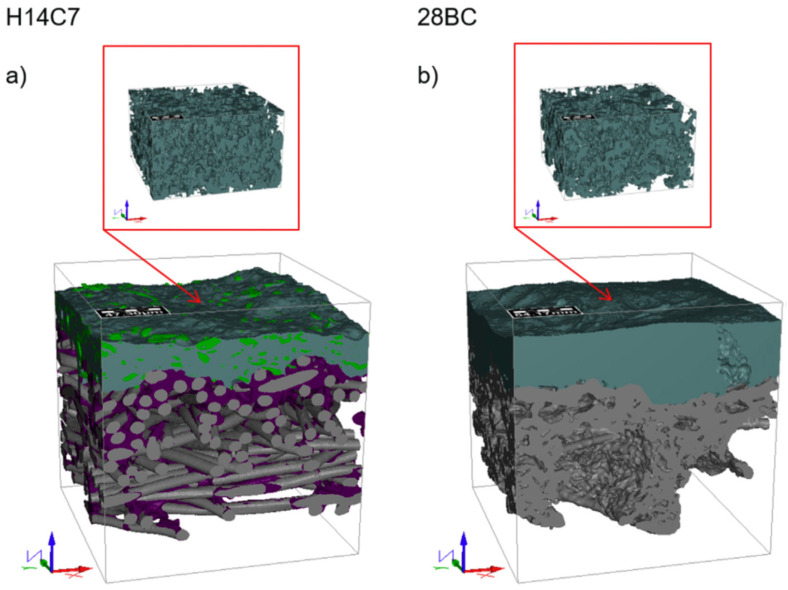
Reconstruction of two GDL microstructures, H14C7 (**a**) and 28BC (**b**), obtained by synchrotron-based X-ray tomography (pixel size: 0.325 μm, while the microporous layer (MPL) in the inlet is the FIB-SEM reconstruction (pixel size: 5 nm; 500 × 300 × 300 voxels) [70].

**Figure 8 materials-16-05808-f008:**
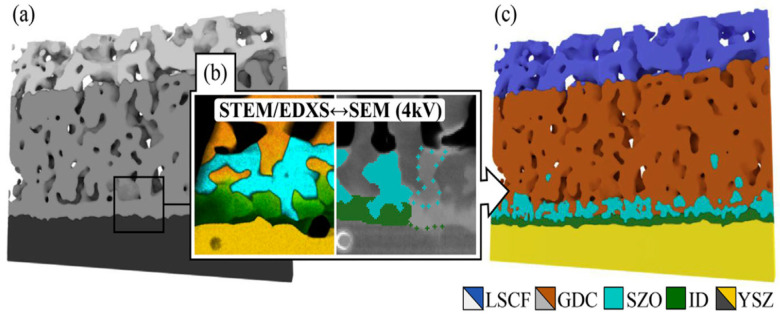
Schematic representation of correlative method proposed by Wankmüller [80]: (**a**) FIB-SEM reconstruction; (**b**) STEM/EDX—SEM correlated image; (**c**) final reconstruction with the elemental identification.

**Figure 9 materials-16-05808-f009:**
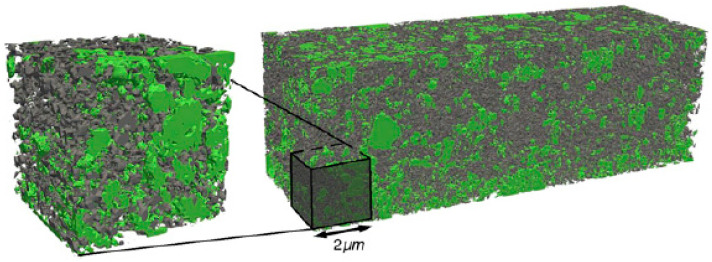
Three-dimensional reconstruction of a composite LiFePO_4_ cathode with three phases, LiFePO_4_ (green), carbon black (black), and pores (transparent) [101].

**Figure 10 materials-16-05808-f010:**
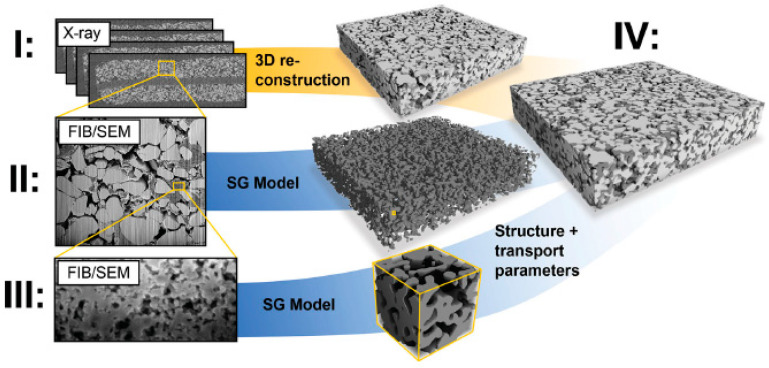
The complete workflow as proposed by Zielke [105] for the reconstruction of a LiCoO_2_ cathode: (**I**) reconstruction of the 3D distribution for the micrometric active material by X-ray tomography; (**II**) reconstruction of the 3D distribution of the carbon binder by FIB-SEM tomography; (**III**) high-resolution FIB-SEM tomography for the observation of the nanoscale features of the carbon binder. (**IV**) Definition of the complete model for a precise reconstruction of the entire system and an overall prediction of the electrode performance.

**Figure 11 materials-16-05808-f011:**
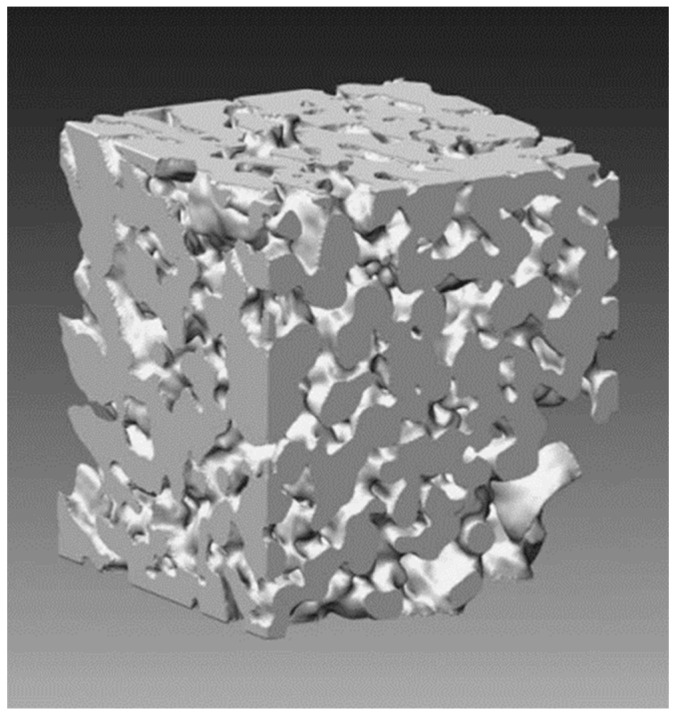
Three-dimensional reconstruction of porous BaTiO_3_ as conducted by Holzer et al. (cube dimensions: 1646 × 1829 × 1743 nm) [14].

**Figure 12 materials-16-05808-f012:**
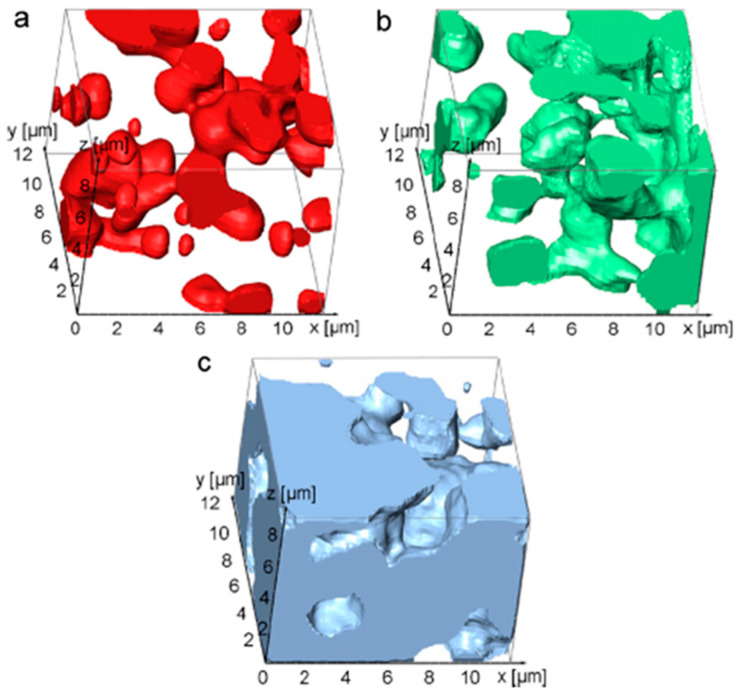
Three-dimensional elemental distribution of calcium (**a**), magnesium (**b**), and titanium (**c**) from the work of Schaffer [130].

**Figure 13 materials-16-05808-f013:**
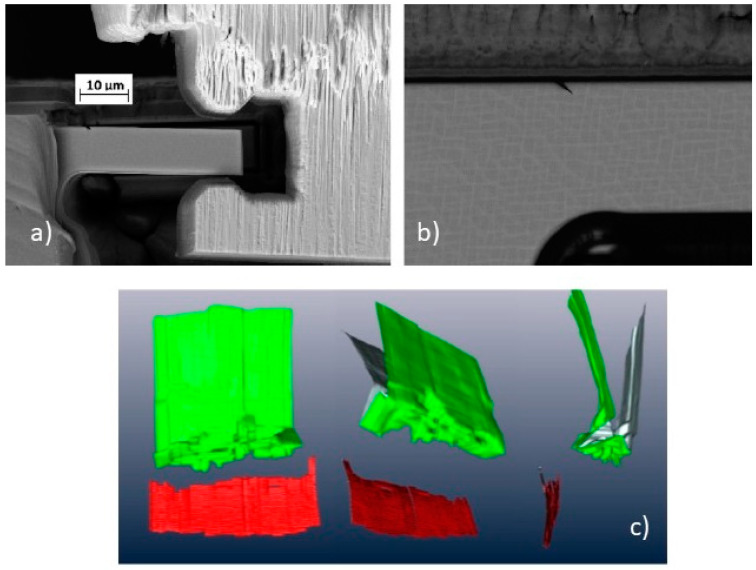
(**a**) Overview of the experiment setup where the machined single-crystal micro-bram and the gripper are exposed; (**b**) detail of the notch before the fatigue experiment; (**c**) FIB 3D reconstruction of the fatigue crack after 6100 load cycle [146].

**Figure 14 materials-16-05808-f014:**
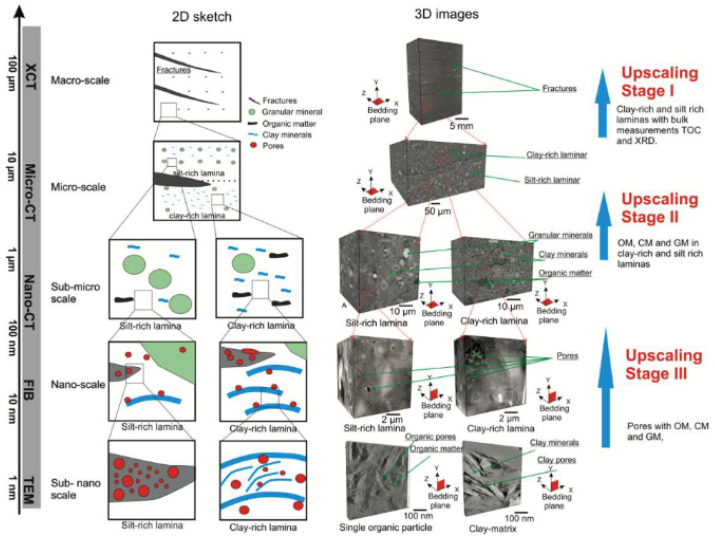
Example of the multiscale workflow for the porosity reconstruction of a shale gas rock, as proposed by Ma et al. [172].

**Figure 15 materials-16-05808-f015:**
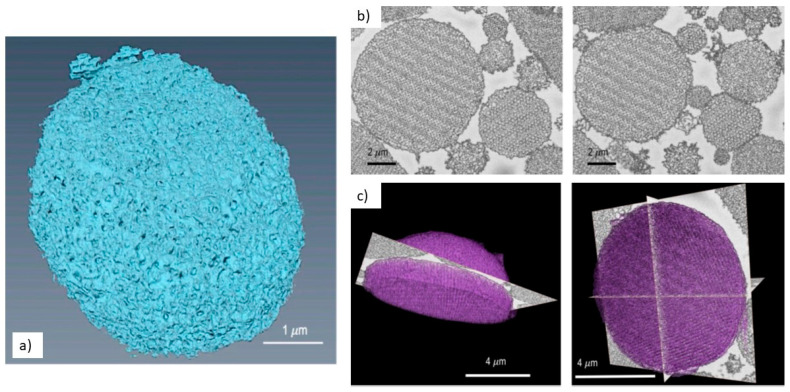
Single porous particles reconstructed by FIB-SEM tomography (**a**) and serial block face (**c**), while in (**b**) are shown the SEM images of two consecutive slices made by the SBS [213].

**Table 1 materials-16-05808-t001:** List of papers on the application of FIB-SEM tomography in battery reconstruction.

First Author (Reference)	Material	Voxel Resolution or Dimensions of the Reconstructed Volume	Associated Techniques	Features of the Rock
Ender [102]	LiFePO_4_ cathode	18.15 µm × 17.75 µm × 27.8 μm	-	Particle size distribution, porosity, active surface area
Eswara-Moorthy [103]	Carbon-based electrode	10 nm × 10 nm × 10 nm	-	Porosity, tortuosity
Wieser [104]	Lithium-ion battery electrode coatings	5 nm × 6.27 nm × 10 nm	Synchrotron radiation computed tomography	Particle size distribution, diffusivity
Prill [32]	Carbon-based negative Li-ion battery electrode	5 nm × 6.27 nm × 10 nm	-	Porosity
Zielke [105]	LiCoO_2_ electrodes	-	Synchrotron radiation computed tomography	Pore size distribution, active area, tortuosity
Etiemble [106]	LiNi_1/3_Mn_1/3_Co_1/3_O_2_ (NMC), LiFePO_4_ (LFP), and blended MC/LFP electrodes	10 nm × 10 nm × 10 nm	Synchrotron radiation computed tomography, BET	Pore size distribution, active area
Besnard [107]	LiNi_1/3_Mn_1/3_Co_1/3_O_2_ (NMC), LiFePO_4_ (LFP), and blended MC/LFP electrodes	10 nm × 10 nm × 10 nm	Synchrotron radiation computed tomography, BET	Pore size distribution, active area, tortuosity
Cadiou [108]	Composite electrode made of LiNi_1/3_Mn_1/3_Co_1/3_O_2_ (NMC), LiFePO_4_ (LFP), carbon black, poly(vinylidene fluoride)	10 nm × 10 nm × 10 nm	XRCT	Volume fraction, porosity, interconnectivity, tortuosity
Vierrath [109]	Carbon binder domain (CBD) of a LiCoO_2_	3.9 µm × 5 µm × 2.3 µm	-	Pore size distribution, tortuosity
Almar [110]	Electrodes extracted from two high-power and two high-energy Li-ion battery consumer cells	Sample A: 16.8 µm × 25.9 µm × 44.9 µm; sample B: 24.0 µm × 25.0 µm × 44.4 µm; sample C: 30.0 µm × 21.6 µm × 32.0 µm; sample D: 32.6 µm × 20.8 µm × 41 µm	-	Porosity, particle size, surface area, tortuosity
Liu [111]	LiCoO_2_ cathode from a commercial cylindrical 18,650 energy cell	35,000 μm^3^	-	Volume fraction, surface area density, feature size distribution, connectivity, tortuosity
Biton [112]	LiFePO_4_ cathodes	10 nm × 10 nm × 15 nm	Optical microscopy, micro CT	Particle size distribution
Song [113]	Pristine and long-term cycled cathodes containing Li(Li_0.2_Mn_0.54_Ni_0.13_Co_0.13_)O_2_	50 nm × 50 nm × 50 nm	-	Particle/pore size distribution
Scipioni [114]	Li-ion battery LiFePO_4_/Carbon black (LFP/CB) cathodes	10 µm × 10 µm × 10 µm	TEM	Particle/pore size distribution, connectivity
Etiemble [115]	Silicon/carbon/carboxymethylcellulose electrode for Li-ion batteries	4 samples: 1760 µm^3^	-	Volume fraction, particle/pore size distribution, connectivity, tortuosity
Scipioni [116]	Commercial LiFePO_4_/ graphite 26,650 cylindrical cell	8 datasets with variable voxel resolution	Micro CT, XPS, XRD	Volume fraction, pore size
Liu [117]	Pristine and cycled LiNi_x_Mn_y_Co_1−x−y_O_2_ (NMC) and Li(Li_0.2_Ni_0.13_Mn_0.54_Co_0.13_)O_2_ (HE-NMC) cathodes	Sample A (NMC): 5100 µm^3^; sample B (HE-NMC): 11,800 µm^3^	-	Particle size distribution, connectivity
Moroni [118]	Lithium manganese oxide composite cathode	31.3 μm × 34.5 μm × 16.8 μm	Synchrotron radiation computed tomography	Pore/grain size distribution
Danner [119]	Ag electrode	27.8 nm × 23.9 nm × 10.0 nm	-	-
Biton [120]	Single Zn dendrite	-	Optical microscopy, micro CT	Coordination number
Yufit [121]	Zn anode tip with dendrites	40 nm × 40 nm × 40 nm	Micro CT, EBSD	Volume and surface of the dendrites
Prill [122]	Nanoporous carbon-based electrodes for electric double-layer capacitors	Sample A: 3.57 nm × 3.62 nm × 10 nm; sample B: 2.38 nm × 2.41 nm × 6.67 nm	-	Porosity
Lagadec [123]	Polyethylene and polypropylene separators	420 µm^3^	-	Porosity, pore elongation
Malik [124]	Silicon–graphene hybrid materials negative electrode in lithium-ion batteries	20 µm × 10 µm × 15 µm	-	Porosity, surface area, tortuosity

**Table 2 materials-16-05808-t002:** List of papers involving FIB-SEM tomography application in solar cells.

First Author (Reference)	Material	Voxel Resolution or Dimensions of the Reconstructed Volume	Associated Techniques	Features of the Rock
Wollschläger [125]	Porous TiO_2_ layers infiltrated with ruthenium molecular sensitizer for DSSC	2.9 nm × 3.7 nm × 30 nm	AFM, TEM, transmission Kikuchi diffraction (TKD)	Particle size and shape, porosity, active surface area
Suter [126]	A: 650 nm thick hematite (α-Fe_2_O_3_) photoelectrode; B: 7570 nm thick lanthanum titanium oxynitride (LaTiO_2_N)	Sample A: 1 µm × 5 µm × 24 µm; sample B: 8 µm × 8 µm × 31 µm	-	Solid phase material distribution, particle/pore size distribution, surface area
Andrzejczuk [127]	TiO_2_ nanotubes	2.24 µm × 1.68 µm × 0.46 µm	TEM tomography	Porosity, morphological parameters of the nanotubes

**Table 3 materials-16-05808-t003:** List of papers involving FIB-SEM tomography in shale rock reconstruction.

First Author (Reference)	Shale Location	Voxel Resolution or Dimensions of the Reconstructed Volume	Associated Techniques	Features of the Rock
Sondergeld [171]	Barnett shale, Texas (USA)	5 × 5 × 2.5 µm	EDS, TEM, Scanning Acoustic Microscopy (SAM), Mercury Injection Capillary Pressure (MCIP), NMR	Pore volume distribution
Peng [173]	Barnett shale, Texas (USA)	-	Micro-CT, Helium Ion Microscopy (HIM)	Pore volume distribution, total organic carbon (TOC), permeability
Roshan [174]	Perth Basin (Australia)	9.2 × 3.8 × 0.5 µm	Micro-CT, NMR, gas porosimetry, Heat Technique Route (HTR)	Porosity
Kelly [36]	Unknown location	Sample A: 5 × 5 × 7 nm/pixel; sample B: 4 × 4 × 5 nm/pixel; sample C: 15.6 × 15, 6 × 10 nm/pixel	Broad Ion Beam (BIB) SEM	Pore volume, permeability
Bai [175]	Utica shale, Quebec (Canada)	9.9 × 9.1 × 10 µm	XRD, EDS	Porosity
Nia [176]	Monterey shale, California (USA)	30 × 30 × 30 nm	MCIP, BET, MICRO-CT	Porosity, permeability
Leu [177]	Jordan shale	10 × 10 × 10 nm	MICRO-CT, Small-Angle X-ray Scattering (SAXS), Wide-Angle X-ray Scattering (WAXS)	Porosity
Tang [178]	Silurian Longmaxi formation, Sichuan (China)	-	NANO-CT, EDS	Porosity
Sun [179]	Silurian Longmaxi formation, Sichuan (China)	14.3 × 14.3 × 14.3 nm	MICRO-CT, XRD	Porosity, permeability, tortuosity
Sun [179]	Wenchang formation, Hoizhou Sag (China)	14.3 × 14.3 × 14.3 nm	MICRO-CT, XRD	Porosity, permeability, tortuosity
Saif [180]	Eocene Green River, Uinta Basin (USA)	14.6 × 18.3 × 10 nm	MICRO-CT, Modular Automated Processing System (MAPS) mineralogy	Porosity
Zhang [181]	Upper Carboniferous Taiyuan (China)	-	NANO-CT, helium pycnometry, high-pressure mercury, low-pressure gas adsorption	Porosity, TOC, microfractures
Ma [182]	Lublin Basin (Poland)	Low resolution: 10 × 10 × 200 nm/pixel; High resolution: 5 × 5 × 20 nm/pixel	MICRO-CT, organic petrology, XRD, BET	Porosity, TOC, connectivity
Ma [181]	Baltic Basin (Lithuania)	Low resolution: 10 × 10 × 200 nm/pixel; High resolution: 5 × 5 × 20 nm/pixel	MICRO-CT, organic petrology, XRD, BET	Porosity, TOC, connectivity
He [183]	Sichuan Basin (China)	100 × 100 × 100 nm	XRD, helium porosity, pulse permeability, MICRO-CT	Porosity, TOC
Wang [184]	Unknown location	3.08 × 3.08 × 3.08 nm/pixel	MICRO-CT	Porosity
Ma [172]	Hayness Bossier shale, Texas (USA)	Resolution: 10 × 10 × 20 nm/pixel; Volume: 8 × 8 × 6 µm	MACRO-CT, MICRO-CT, NANO-CT, TEM Tomography, EDX, XRD	Porosity, permeability, TOC, connectivity
Chen [185]	Wufang Longmaxi shale, Sichuan Basin (China)	-	NMR, MICRO-CT, nitrogen adsorption, CO_2_ adsorption, mercury intrusion porosimetry	Porosity, water filling and water removal mechanism
Zhang [186]	Permian Basin, Texas (USA)	Resolution: 5 × 5 × 5 nm/pixel; Volume: 15 × 10 × 5 µm	MICRO-CT, EDS	Porosity, diffusivity
Goral [57]	Vaca Muerte shale (Argentina)	Resolution: 2.5 × 2.5 × 5 nm/pixel; Volume: 5 × 4 × 3 µm	EDS	Porosity, connectivity
Goral [187]	Marcellus shale, New York (USA)	5 × 5 × 5 nm/pixel		Porosity, permeability
Goral [188]	Mancos shale (USA)	Resolution: 10 × 10 × 10 nm/pixel; Volume: 10 × 14 × 10 µm	MICRO-CT, light microscopy	Porosity

## Data Availability

No new data were created or analyzed in this study. Data sharing is not applicable to this article.

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
