# Peer review of "Advances in Focused Ion Beam Tomography for Three-Dimensional Characterization in Materials Science"

_materials, 2023, doi:10.3390/ma16175808_

Round 1

Reviewer 1 Report

The work “Advances in Focused Ion Beam Tomography for Three-Dimensional Characterization in Materials Science” by Mura and cowokers is a review based on one type of tomorgraphy technique for materials.

Always in reviews you could ask for more and more, and reformulate, but I am really not capable of that and I would only ask to do 1 additional paragraph of SEM and TEM because it really is the basis of everything. And to cite some work, which in fact, have surely already been cited afterwards. So, I would say it is the first and maybe the last time it will happen to me. I have to say that the reference list is more than extensive, it is very comprehensive, but I cannot accept the article because I found it surreal that there was no self-citation, from any of the authors. I would agree, but I cannot. Avoiding self-quotes is always good, but it is sometimes an exaggeration, and you also need to contextualize it because you are writing an article and even a review...

There is some confusion sometimes with some term that probably comes from typos: ALIGMENT or ALIGNMENT, or are different meanings? However, there are few errors that can be solved in the PROOFS stage.

I have to say that the review is good for getting a clear picture of how the technique is applied to materials, and at least in my case, it was an easy read. Thus, I recommend the paper for publication in MATERIALS.

Author Response

We thank the reviewers for their interesting suggestions.

1) We had just added a simple sentence about TEM sample preparation in the introduction, in addition to the description of TEM tomography, because this article is about a specific methodology for a particular class of scanning electron microscopes called dual beam. So, in our view, a whole paragraph on TEM can be confusing. About SEM, we have improved Figure 2 and its description, and a lot of information about its operation has already been given in sections 2.1, 2.2 and 2.3.  

2) In the introduction we have added two articles by prof. Ferroni and dr. Morandi on an innovative system for operating STEM tomography inside a SEM.

3) We have corrected the word alignment and made a thorough revision of the text.

Reviewer 2 Report

In this review paper, the authors summarized FIB-SEM tomography techniques for the detection of three dimensional reconstruction of microscopic structures and showed that this method is routine tool for the material characterization. However, before it has the possiblity to be published, there are some comments should be well concerned by authors. 

1. The basic mechanism of FIB-SEM tomography should be clearly addressed in the section of 2.1.

2. This paper lacks of outlook analysis of this method in the final section.

3. The English of this paper should be well polished.

Author Response

We thank the reviewer for his\her suggestions.

We have done a deep revision of the entire text, including an improvment of section 2.1, 2.2, 2.3 (for the FIB-SEM functioning) and the final conclusions.

Reviewer 3 Report

In the present paper, the authors have provided an insightful and comprehensive overview of the crucial technique of FIB-SEM tomography, which holds immense significance for the three-dimensional reconstruction of microscopic structures with nanometric resolution. The paper eloquently covers the entire process, starting from the experimental setup, followed by data analysis, and culminating in the final reconstruction. Such a meticulous description ensures that readers, both experts and newcomers to the field, can grasp the intricacies involved in this cutting-edge imaging approach. The versatility of FIB-SEM tomography is amply demonstrated through an extensive list of applications, encompassing diverse fields from batteries and shale rocks to various types of soft materials. This breadth of applications underscores the technique's universal applicability, making it an indispensable tool for researchers across different disciplines. Moreover, the paper adeptly highlights the continuous technological and algorithmic advancements in the field. These developments not only enhance the resolution capabilities but also significantly expand the degree of automation and compress the overall duration of the process. The amalgamation of these advancements with the rapid progress in artificial intelligence promises to further propel FIB-SEM tomography into a routine tool for material characterization in the near future. After thorough consideration, I believe that this paper holds substantial value for the readers of this esteemed journal, especially with some recommended revisions to enhance its impact.

1. The Abstract should be modified to include a succinct statement regarding the future outlook of FIB-SEM tomography. Mentioning the anticipated direction of research and potential breakthroughs in the field will enrich the Abstract and set the stage for readers' expectations.

2. While the figures in the paper are generally illustrative, an improvement in the quality of Figures 1 and 2 would enhance the visual clarity for readers, allowing them to better grasp the nuances of the experimental setup and data analysis processes. 

3. To ensure academic rigor, it is imperative that all the presented equations in the paper are accompanied by appropriate references. This step will provide readers with a reliable source to delve deeper into the mathematical foundations of the techniques involved.

4. To aid readers in gaining a more comprehensive understanding of the applications of FIB-SEM tomography in batteries and solar cells, the authors are encouraged to create a comparative table similar to Table 1. This table can serve as a quick reference guide, providing a clear overview of the strengths and limitations of the technique when applied to these specific materials.

5. In the conclusion section, the authors should take the opportunity to delve into greater detail about the future prospects and potential developments in the field of FIB-SEM tomography. This will offer readers valuable insights into the promising avenues for future research and technological advancements.

In conclusion, with the incorporation of these recommended revisions, the paper will undoubtedly be an essential resource for researchers, academics, and professionals alike, furthering our collective understanding of FIB-SEM tomography and its burgeoning role in material characterization and beyond.

Author Response

We thank the reviewer for his\her suggestions.

1) We have written a new version of the abstract following the suggestion of the reviewer.

2) We have made a new version of Fig. 2, improved with a more detailed description of the system. Fig. 1 is just a basic image showing the characteristics of FIB-SEM tomography compared to other tomographic techniques. We have already sent a graphical abstract with an image illustrating all the steps required in FIB tomography.

3) As suggested by the reviewer, we have added the requested reference in the new version of the article.

4) As suggested by the reviewer, we have added the two requested tables in the new version of the article

5) We have written a new version of the conclusion following the suggestion of the referee

Reviewer 4 Report

Manuscript is written well. It covers a very good topic. As the topic is new some recent references can also be added (2021-2023).

Additionally, if the author would like to add some literature relate to RAM , mechanical failure of devices, it will make it more comprehensive.

Author Response

We thank the reviewer for his\her comment about our review.

In response to his request, the reviewer will find some articles on the application of FIB-SEM to failure mechanisms in section 3.5, where we have compiled papers about steel, metals and alloys. We didn't go into detail on failure mechanisms because the materials studied in these papers, which are included in this section of the review, are very different and this could affect the readability of the article dedicated to FIB-SEM tomography.
